# Delivering screening programmes in primary care: protocol for a scoping and systematic mixed studies review

Rakesh Narendra Modi ,[1] Sarah Kelly,[2] Sarah Hoare ,[2] Alison Powell,[2] Isla Kuhn,[3] Juliet Usher-Smith ,[1] Jonathan Mant,[1] Jenni Burt[2]

[1]Primary Care Unit, University of Cambridge, Cambridge, UK
[2]he Healthcare Improvement Studies Institute, University of Cambridge, Cambridge, UK
[3]Medical Library, School of Clinical Medicine, University of Cambridge, Cambridge, UK

**Correspondence to**
Dr Rakesh Narendra Modi;
rnm30@medschl.cam.ac.uk

## ABSTRACT

**Introduction** Screening programmes represent a considerable amount of healthcare activity. As complex interventions, they require careful delivery to generate net benefit. Much screening work occurs in primary care. Despite intensive study of intervention delivery in primary care, there is currently no synthesis of the delivery of screening programmes in this setting. The purpose of this review is to describe and critically evaluate the delivery of screening programmes in general practice and community services.

**Methods and analysis** We will use scoping review methods to explore which components of screening programmes are delivered in primary care and systematic review methods to locate and synthesise evidence on how screening programmes can be delivered in primary care, including barriers, facilitators and strategies. We will include empirical studies of any design which consider screening programmes in high-income countries, based in part or whole in primary care. We will search 20 information sources from 1 January 2000, including those relating to health (eg, MEDLINE, Embase, CINAHL), management (eg, Rx for change database) and grey literature (eg, OpenGrey, screening committee websites). Two reviewers will screen citations and full texts of potentially eligible studies and assess these against inclusion criteria. Qualitative and quantitative data will be extracted in duplicate and synthesised using a best fit framework approach. Within the systematic review, the mixed methods appraisal tool will be used to assess risk of bias.

**Ethics and dissemination** No ethics approval is required. We will disseminate findings to academics through publication and presentation, to decision-makers through national screening bodies, to practitioners through professional bodies, and to the public through social media.

**PROSPERO registration number** CRD42020215420.

## INTRODUCTION
### Background

Screening aims to identify people with or at risk of a particular condition in an apparently asymptomatic population, with the goal of reducing the risk of ill health related to that condition.[1–3] Official screening programmes, such as breast cancer screening or newborn

### Strengths and limitations of this study

► A scoping review followed by a systematic review will together consider a broad spectrum of issues in the delivery of screening programmes in primary care.
► The search of 20 databases and additional key publications will provide a comprehensive basis on which to identify relevant evidence.
► The integration of both quantitative and qualitative data will generate a conceptualisation of screening delivery from multiple paradigms.
► The use of a best fit framework approach will provide accessible answers to the review questions in a diagrammatic format.
► Resource limitations mean that only English language publications have been included in this review.

bloodspot testing, account for considerable health spending and activity in high-income countries: for example, in 2015, the UK National Health Service performed 21 million official screening tests in a population of 65 million.[4–7] These screening programmes typically receive support from the public, professionals and decision-makers, stimulated by the rationale that early detection may prevent ill health.[3 8–10]

Despite positive views, numerous harms of screening are recognised. The identification of participants as having a condition they do not (false positives) and the failed identification of participants who do have the condition (false negatives) is one major source of harm. Harm can also arise through identification of conditions that may never cause illness (overdiagnosis) or treatment of conditions that may never change the outcome (overtreatment).[3 9–12] Screening programmes actively approach people who have not sought medical help, which may have psychosocial consequences.[3 6] The effective and efficient delivery of screening programmes seeks to

maximise benefits (including cost benefits) and minimise harms.[3 7 13]

The delivery of complex healthcare interventions, such as screening, has been the study of multiple disciplines under terms such as 'implementation science' and 'translational research'.[14–16] A number of theories, models and frameworks have been developed to explain and guide the successful delivery of interventions in order to create impact at a reasonable cost.[14 17–19] Initial relatively simple theories of implementation[17 20] have given way to progressively more complex models taking into account multiple layers of agents, networks and dynamics.[14 17 21 22] In parallel, there have been increasing numbers of empirical studies of the implementation of healthcare interventions, and related guidance on how best to design such studies.[14 18 19 21 23–25] To date, however, there has been no synthesis of these studies relevant to the delivery of screening programmes in general.

Primary care, including general practice and community care services, is central to the delivery of many screening programmes (eg, cervical screening, general health checks),[13 26–31] reflecting its prominent health system role in early detection and disease prevention.[28 32–35] Primary care professionals are familiar with the explanation of results and referrals to specialist care required following screening and the holistic management of illnesses that might be diagnosed through screening.[29 30 33] Health systems that use primary care to greater degrees tend to be more cost-effective, and many health systems are further developing the role and nature of primary and community services.[36] As interest in new screening approaches continues to evolve, an expanded role for primary care can be anticipated. However, any new programmes may place additional strain on an already stretched system,[37 38] and evidence on the effective implementation of screening in primary care is required to ensure efficient use of resources.[18]

This review will first scope the literature to consider which aspects of screening programmes have been delivered or could be delivered, by general practice and community care services. It will then explore how such screening programmes have been delivered in primary care and the consequences of varying approaches to delivery. Syntheses of evidence will be used to generate recommendations for policy-makers, health professionals and researchers of screening programmes. The term 'general practice' is synonymous with family practice and family medicine,[39] and the term 'primary care' refers to general practice and community services.

## Objectives

We will review the literature to describe and critically evaluate the delivery of screening programmes in general practice and community services.

We will conduct two linked reviews, answering the following questions:

1. What screening programmes or components of screening programmes have been or could be delivered in general practice and/or community services?
2. How have general practices undertaken the delivery of screening programmes or components of screening programmes, and what has happened as a result of different approaches to delivery?

With question one, our primary objective is to produce a definitive summary of the screening programmes or components of screening programmes that general practices and/or community services have delivered, are delivering or have the potential to deliver. We have included both general practices and community services due to global trends towards integration.[40–43] This will provide important background context for question two, with a primary objective of locating, critically appraising and synthesising the evidence on approaches to the delivery of screening programmes specifically in general practices, including a consideration of the varied systems and strategies used and the barriers, facilitators and outcomes of these. These findings will enable us to generate recommendations for the delivery of screening programmes in general practice. Community services were not included in the scope of question two as we felt that the influences on delivery would be too different to those of general practice to provide any meaningful recommendations.

In addition, our focus will be on screening for primary prevention as these mostly involve larger systems, greater coordination and affect more of the population.

## METHODS AND ANALYSIS

We will conduct a systematic scoping review to answer question one as the aim is simply to map screening activities that have occurred in primary care.[44] We will conduct a systematic review, including risk of bias assessment, to answer question two.[45] In both reviews, we anticipate locating relevant qualitative and quantitative data: we will use a best fit framework approach to guide data synthesis (see below).[46–48]

### Conduct and reporting

The scoping review for question one will be conducted and reported in accordance with the Preferred Reporting Items for Systematic Reviews and Meta-Analyses (PRISMA)-Scoping Review (ScR) statement.[44] The systematic review for question two will be conducted and reported in accordance with the PRISMA statement.[49]

This protocol is based on the PRISMA-P guidance[50] and its PROSPERO[51] registration number is CRD42020215420.

### Eligibility criteria

Due to the broad scope of these reviews, eligibility criteria have been developed iteratively through the conduct of preliminary searches, examination of records returned and discussions among the authors about areas of contention.

## Population

We will include studies that involve primary care institutions, staff, their systems or their actions (eg, policies, communications etc). For question one, these will include community services such as pharmacies and outreach programmes. For question two, these will be limited to general practices.

## Intervention

For question one, eligible studies will describe the components of screening programmes that have been implemented in primary care. For question two, studies will describe how screening programmes have been delivered in general practice and/or evaluate the different approaches to delivery. For question two, this will include studies exploring barriers, facilitators, systems, strategies or other features that relate to the delivery of screening programmes or components of screening programmes. External influences such as international policy or patient attitudes may impact on the delivery of screening in general practice; however, studies on such aspects will be included only if they include a specific focus on how these relate to the implementation of screening in general practice. This is to ensure a focus is maintained on components of screening which are under the control of general practice decision-makers.

For both questions, we will focus on screening programmes that are consistent with the general definition that is broadly accepted and stated above: the process of identifying people with or at risk of a particular condition in an apparently asymptomatic population, for the purposes of reducing the risk of ill health related to that condition.[1–3] The key feature is that screening is performed for the purpose of preventing morbidity or mortality. As such, we will only examine those programmes for which there is a clear system for managing the patient following results, from further tests to treatment and follow-up. Polygenic risk scores and new directly marketed screening tests, which provide only an estimate of future disease risk and no clear management plan, will be excluded.[3 52] We will also only include screening programmes that involve systematic invitations to an eligible population and we will exclude programmes that solely rely on opportunistic invitations.[3] We will only study screening programmes for purposes of primary prevention.

## Comparator

No comparator group is required.

## Phenomena and outcomes

For question one, included studies will describe the type of screening programmes or components of these screening programmes delivered within primary care. Question two is concerned with exploring how screening has been delivered in general practice and what the results of different approaches to delivery have been. Potential phenomena and outcomes of interest include the systems and strategies used to deliver screening, barriers and facilitators of

such approaches and why such approaches were taken. Some studies will also report impacts of these phenomena and outcomes. These are not essential to be included and could be clinical (eg, changes in mortality) or implementation impacts (eg, changes in attitudes, reach or maintenance).[19 24]

## Types of studies

Empirical studies that involve primary data collection or secondary analyses of primary data will be included. We will include all study designs, whether interventional (including randomised controlled trials and quasi-experimental designs), observational (including cohort studies, case–control studies, cross-sectional studies), qualitative, mixed methods, case reports or case studies. Articles do not need to be peer reviewed, for example, conference abstracts or grey literature. Studies that do not involve primary empirical research (eg, reviews, editorials) will be excluded; if found during our search, their reference lists will examined for eligible primary studies. Letters may be included if they contain a primary study.

## Setting

Included studies must relate to a primary care setting. In high-income countries, which are of interest to this review, this can be categorised into general practice (health professionals in a local health facility that include family practitioners, who provide some continuous care for all conditions) and community services (other services that provide local health services such as pharmacies and community nursing).

For question one, both general practice and community services are of interest as they are undergoing integration and it will be useful to note their potential in delivering screening programmes. It will also enable us to synthesise a map of screening activities that differentiates their activities. For question two, only general practice will be of interest because issues that affect delivery in general practice were felt likely to be too different to those in community services and therefore a combined synthesis may not provide useful recommendations.

## Report characteristics

Studies published from 1 January 2000 onwards will be included to reflect the current role of primary care in the delivery of screening programmes.

## Exclusion criteria

► Studies not published in English.
► Secondary research such as reviews and editorials.
► Unpublished material such as personal communications and drafts.
► Publications that discuss screening delivery but do not clearly base their discussion on primary empirical findings.
► Studies of screening programmes that do not have a clear management plan for results, for example, those that use polygenetic risk scores to provide only an estimate of future disease risk and do not have a clear

**Table 1** Information sources for the review of the delivery of screening programmes in primary care

| Databases | Grey literature sources | Other sources |
|---|---|---|
| MEDLINE via Ovid | UK NSC website | Author recommended papers |
| Embase via Ovid | PHE screening blog | UK NSC recommended papers |
| CINAHL via EBSCOhost | National Health Service (NHS) health check website | Reference lists of relevant reviews |
| Scopus | USPSTF website | |
| ASSIA via Proquest | OpenGrey | |
| PsycINFO via EBSCOhost | OpenSIGLE | |
| | Google Scholar | |
| | NICE Evidence | |
| | ICTRP | |
| | Clinicaltrials.gov | |
| | Rx for change database | |

ASSIA, Applied Social Sciences Index and Abstracts; CINAHL, Cummulative Index of Nursing and Allied Health; ICTRP, WHO International Clinical Trials Registry Platform; NICE, National Institute for Health and Care Excellence; UK NSC, UK National Screening Committee; PHE, Public Health England; USPSTF, United States Preventive Services Task Force.

management plan. Screening for genetic disorders where there is clear management of the patient after results are included, for example, BRCA 1 or 2.

► Studies on screening for secondary or tertiary prevention where a patient is invited because he/she has a disease that predisposes to the condition that he/she is being screened for, for example, diabetic retinopathy screening.

► Studies where the participants were not systematically invited for screening, such as opportunistic screening programmes and commercial genetic screening tests.

► Studies on screening delivery in settings other than primary care, such as hospitals or central public health institutions.

► Studies with all data based in low-income or middle-income countries, according to the World Bank.[43 53 54]

### Information sources
#### Databases
Table 1 displays the information sources that will be used in their broadest dates of coverage; these were searched in September to November 2020. These sources were chosen after discussions with an information specialist, the review team and members of the UK National Screening Committee.

Other sources of publications (see table 1) will be recommended by the review authors through their experience of research and practice in health screening and through their initial examination of the results from scoping searches. We have emailed known experts in the UK National Screening Committee, who have published on screening and its implementation.[3] Snowball email contact with further experts suggested by the initial experts will also be performed as appropriate. We will also employ screening of reference lists of relevant review articles that are retrieved by our search in order to find further eligible primary studies.

### Search strategy
A draft search strategy was created by an information specialist (IK), a general practitioner (RNM), a health services researcher (JB) and a reviews specialist (SK) with contributions of terms by all authors. Searches were piloted in all research databases employed in this review, records were examined, key papers were checked for their presence and through this process the strategy was iteratively refined. The concepts 'screening', 'primary care' and 'implementation' were combined with 'and' operators. Each concept consisted of controlled vocabulary terms (eg, Medical Subject Headings - MeSH terms) and free-text terms that were combined with 'or' operators. The terms will be translated for each database including, where available, the use of controlled vocabulary. The same search strategy will be used for both questions one and two. A draft of this in the format for Ovid MEDLINE is shown in online supplemental file 1.

### Study records
#### Data management
We will use Covidence[55] to store and manage records returned from main searches. Other software used will include Endnote,[56] NVivo,[57] Microsoft Word, Microsoft Excel and Microsoft PowerPoint.

#### Selection process
Records will be downloaded into Covidence for screening. Titles and abstracts of the records returned from the searches, as well as additional publications identified through other sources, will be screened (blinded and in duplicate) by independent reviewers according to the eligibility criteria for both questions one and two. If disagreements between the reviewers cannot be resolved through discussion, a third reviewer will arbitrate the final decision. The full texts of the remaining records will be screened for potential relevance, blinded and in duplicate, according to the same eligibility criteria. Where there is disagreement despite discussion, a third reviewer will arbitrate.

### Data and phenomena/outcome items
Although data in this mixed studies review are likely to include unforeseen categories, data extracted for all studies will include:

- ► Study characteristics: reference, first author, year, funding, country from which data were collected, sample size, intervention/purpose of the study, health condition screened for, primary care staff involved, comparator, data collection method and study category according to the mixed methods appraisal tool (MMAT)[58]—'qualitative', 'quantitative randomised controlled trials', 'quantitative non-randomised', 'quantitative descriptive' or 'mixed methods'.
- ► Key demographics, where relevant, for example, age, health status and socioeconomic status.
- ► Key contextual factors, where relevant, for example, new policies, health emergencies, response to significant events, economic concerns and particular health systems.

Data will be extracted into a Microsoft Excel spreadsheet by two independent reviewers, blinded and in duplicate. Disagreements will be discussed by the two reviewers and arbitrated by a third reviewer if necessary. Data relevant to the review questions will be extracted in a similar way and synthesis will then proceed, guided by the best fit framework approach. It is not possible to precisely prespecify the data available for extraction and how it will be categorised for synthesis, but we anticipate potential areas may include:

- ► For question one: setting (general practice, community service), relevant component of a screening programme and related outcome data.
- ► For question two: phenomena measured (eg, perceived barriers, effects of audits), types of outcome (qualitative or quantitative) and outcome data. Qualitative data extracted will be direct quotes from participants, themes generated by the authors where they are clearly supported by data and any quantitative data.

### Risk of bias in individual studies
For studies included in response to question two, the MMAT[58] (see online supplemental file 2) will be used to assess the risk of bias at study and outcome level. This will be done by two independent reviewers, blinded and in duplicate. Disagreements will be resolved by discussion, and where agreement cannot be made, a third reviewer will arbitrate.

### Data synthesis
#### The best fit framework approach
Qualitative and quantitative data will undergo a mixed methods synthesis using a best fit framework approach.[47 48] This approach was developed in response to concerns about the resources required to perform qualitative syntheses using approaches such as framework synthesis, thematic synthesis or meta-ethnography. A framework synthesis may require opinions of the authors that are not always transparent, while thematic synthesis or meta-ethnography may require intensive inductive analysis that can be prohibitively time consuming in large reviews.[48] The best fit framework approach provides a rapid and

transparent method for synthesising qualitative data, by using a coding framework built from relevant theories and models identified in the literature. It therefore facilitates an examination of review findings in the light of pre-existing theories (with the potential for further development of these theories) rather than generating theories de novo. By combining the deductive methods of framework synthesis (by coding data to a priori themes) and inductive methods of meta-ethnography (for data that does not code to a theme in the framework), the best fit framework approach is particularly useful where there are relevant theories in the literature but they have not been refined the particular context of the review question. The method generates a diagrammatic answer to the review question that aims to be easy to understand and clear for policy-makers.

A best fit framework approach to synthesis thus offers a number of advantages for this review. Notably, theories and models of implementation are available in the literature but have not yet been tested and refined with regard to screening in general. We recognise that best fit framework approaches were originally created for qualitative syntheses, and we are employing a reasonably novel approach of using it for a mixed studies review. However, both qualitative and quantitative data may add depth to and refine a synthesis framework, and combining both is readily straightforward when, as here, quantitative data are likely to be too varied to allow a meaningful meta-analysis.

#### Searching for models for the best fit frameworks
In this review, a best fit framework will be generated to guide data synthesis for each research question. To produce the initial frameworks, we will systematically search the literature for existing models of relevance to the review questions. These searches will be separate from the main literature search for the review that has been described above. The models may relate to certain core features of the review questions, but not necessarily to the specific context of interest. If multiple models are found, those that are most relevant will be purposively selected and synthesised. For question one, the search will aim to locate publications relevant to creating a pathway of steps in a screening programme. For question two, the search will aim to locate publications relevant to the delivery of screening programmes. We will generate initial search terms using the Behaviour of interest, Health context, Exclusions, Models or theories mnemonic (BeHEMoTH), created to guide systematic searches for theories.[48 59] We will translate initial terms into suitable search strings for two main databases: EBSCO CINAHL and Ovid MEDLINE (see online supplemental file 3 for Ovid MEDLINE search terms for question one and online supplemental file 4 for question two).

Relevant located models will be chosen using an adaptation of the approach outlined in the original description of the best fit framework method[48] and the BeHEMoTH search strategy.[59] As with the original best fit framework

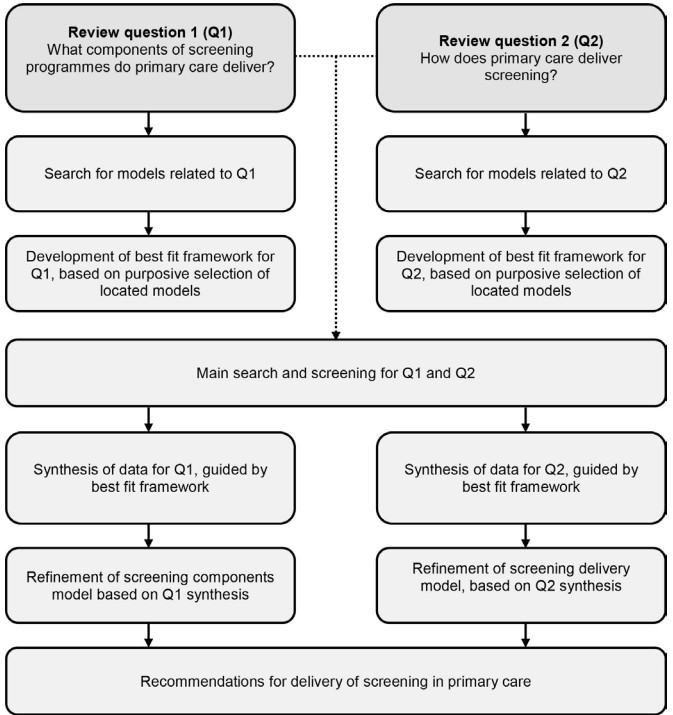

**Figure 1** Explanation of the role of best fit frameworks in the syntheses for the scoping and systematic reviews on the delivery of screening programmes in primary care.

approach, we will conduct systematic searches to locate relevant models. However, we will expand the search with controlled vocabulary terms, add key papers suggested by the research team and we will purposively select final included models to increase variation in concepts and to represent the different levels of delivery in screening programmes (eg, public health management and general practice staff). We will also draw on relevant models known from the grey literature.[48]

## Synthesis using the best fit frameworks

In this way, best fit frameworks will be produced separately for question one and question two. The final frameworks (ie, the outcomes of the searches for models), whether representing a single pre-existing model or a synthesis of more than one pre-existing model, will subsequently be used as a framework for coding data extracted from included studies.[47] Extracted quantitative and qualitative data will be coded against the best fit framework concepts. Relationships between concepts that emerge from the data will also be noted and will help understand how concepts relate to each other. Where there is not an appropriate concept, a new code will be created. This code will either be an original code or an existing code from another model encountered during the search for the best fit framework. These new codes might undergo concurrent thematic analysis, leading to refinement of existing concepts in the best fit framework, the creation of new concepts or the alteration of relationships between them, as appropriate. The concepts of the framework that have become redundant after this process may be removed.[48]

For each review question, this will therefore produce a framework that has been refined by qualitative data and described by quantitative data, which will be narratively synthesised where there is sufficient information. By using the relationships between concepts uncovered in this process, a new model that displays these relationships will be produced to answer each review question. Data, codes, themes and models will be discussed in regular team meetings, which will actively encourage analyses through different perspectives and assumptions, contributing to the robustness of the findings. An overview of the incorporation of the best fit framework approach in this review is shown in figure 1.

The authors of the original best fit framework method recommended conducting a sensitivity analysis by removing low-quality studies.[48] However, this is not appropriate for this review: assessment of the quality of qualitative studies is often only based on the quality of reporting and it is widely recognised that for complex interventions[60] even 'low-quality' studies may provide useful indications of relevant concepts that need further exploration. We will, however, use the risk of bias assessment to inform our interpretation of findings for review question two.

### Meta-biases and confidence in cumulative evidence

Publication bias will not be assessed due to the variety of evidence types in this review and its lesser usefulness in a complex exploratory review such as this. However, by including grey literature and databases that include conference abstracts, we hope to limit this.[61] We will reflect and comment on individual study risk of biases, recurrent types of bias and potential effects of these on the overall conclusions for review question two, including how they might create differences in study findings. The strength of recommendation may be described with a GRADE (Grading of Recommendations, Assessment, Development and Evaluations) Score informed by the MMAT scores of individual studies that address the same recommendation. However, due to the variety of studies that may help answer the review question, such a score may not prove useful; in this case, this process will not be undertaken.

### Patient and public involvement

Patients and public were not involved in the design or writing of this publication as this review will mainly generate lessons for healthcare providers and decision-makers. We have, therefore, included authors in these roles. We do, however, intend to disseminate the results of the review via social media and directly to relevant contacts in patient organisations and primary care bodies.

### ETHICS AND DISSEMINATION

All data are available in the public domain so ethical approval is not required. If major amendments are required, the nature, rationale and date will be

documented in PROSPERO. Outputs will be published in peer-reviewed journals and presented at international conference. Publications will be shared through reviewers' and institutions' social media, networks and websites, thus reaching academics, policy-makers and the public. They will also be sent to members of various national screening committees and primary care bodies through established links.

**Acknowledgements** The authors are grateful to Professor Anne Mackie and Katharine Thompson of Public Health England and Matthew Fagg of the National Health Service (NHS) Diabetes Programme for advising on key grey literature related to screening delivery in primary care.

**Contributors** JB is the guarantor. RNM drafted the manuscript. All authors contributed to eligibility criteria and review methods and edited and approved the final manuscript. RNM, JB, IK and SK created the search strategy. JU-S, SH, AP, JM and JB provided expertise on screening.

**Funding** RNM's work for this review was supported by the Wellcome Trust as part of the Wellcome Trust PhD Programme for Primary Care Clinicians (grant number 203921/Z/16/Z). SK, IK and JB were supported by the Health Foundation's grant to the University of Cambridge for The Healthcare Improvement Studies Institute. SH and AP contributed to this publications through independent research funded by the National Institute for Health Research (NIHR) under its Programme Grants for Applied Research Programme (Reference Number RP-PG-0217-20007). JU-S and JM were funded by the University of Cambridge. JM is an NIHR Senior Investigator. All the funders had no involvement in the development of this protocol and will have no involvement in any aspect of the review itself. The views expressed are those of the author(s) and not necessarily those of the NHS, the Wellcome Trust, the NIHR or the Department of Health.

**Competing interests** RNM, SH, AP, JM and JB are undertaking a trial of Atrial Fibrillation Screening funded by the National Institute for Health Research.

**Patient consent for publication** Not required.

**Provenance and peer review** Not commissioned; externally peer reviewed.

**ORCID iDs**
Rakesh Narendra Modi http://orcid.org/0000-0001-9651-6690
Sarah Hoare http://orcid.org/0000-0002-8933-217X
Juliet Usher-Smith http://orcid.org/0000-0002-8501-2531

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
