## [Reviewer comments · BMJ Open]

ARTICLE DETAILS

TITLE (PROVISIONAL)	Delivering screening programmes in primary care: protocol for a scoping and systematic mixed-studies review
AUTHORS	Modi, Rakesh; Kelly, Sarah; Hoare, Sarah; Powell, Alison; Kuhn, Isla; Usher-Smith, Juliet; Mant, Jonathan; Burt, Jenni

VERSION 1 – REVIEW

REVIEWER	Campbell, Christine University of Edinburgh, Usher Institute
REVIEW RETURNED	05-Jan-2021

GENERAL COMMENTS	This is a clearly written protocol for a systematic literature review of the evidence relating to delivery of screening programmes in primary care. There is currently no published systematic review on this topic and the results will be of wide interest to policy makers, screening programmes in high income settings, and within primary care and community services. The approach includes an initial scoping review followed by a systematic literature review that will adopt best practice systematic review methodology; the protocol has been registered with PROSPERO. The authors acknowledge that use of only English-language literature will be a limitation. The scope is comprehensive with searching of 20 information sources proposed, and findings from both qualitative and quantitative studies will be included, and integrated using a best fit framework approach. This reviewer agrees with the authors' approach of keeping in 'low quality' studies, but it would be helpful to know how they will manage this if there is substantial weakness in the literature for one type of screening or element of delivery. Authors are excluding screening for conditions for secondary or tertiary prevention – this reviewer understands the reason for this (the focus of the protocol is on primary prevention) but I also wonder if any lessons can be learnt from any delivery of these services in primary care. The results will have real potential to inform and influence future delivery of screening, and I look forward to publication of the study findings in due course.
--

REVIEWER	Nascimento, Bruno Universidade Federal de Minas Gerais, Serviço de Cardiologia e Cirurgia Cardiovascular
REVIEW RETURNED	04-Mar-2021

GENERAL COMMENTS	Dear authors, This study protocol is interesting and ambitious, aiming to evaluate screening programs in primary care in general. Some points,
---

	however, deserve additional clarification. Please see my comments: 1) Intro: maybe some examples of screening programs should be provided (e.g. diabetes, hypertension, echo screening for conditions such as structural heart disease and RHD). 2) For research question 2, please clarify if the review is looking for outcomes (e.g. clinical outcomes). If so, they should be better specified, as different types of interventions will be measured by different kinds of outcomes. 3) In the sentence: "Polygenic risk scores and new directly marketed screening tests, which provide only an estimate of future disease risk and no clear management plan, will be excluded" it is also not clear how the inclusion criteria will be applied: will the authors only include studies with a longitudinal design, that explored outcomes resulting from screening? As an example: echo screening studies, aimed at diagnosing heart disease in high-risk populations, will be excluded? Is that true for research questions 1 and 2 or only for research question 2? Please clarify. 4) Do the authors plan a sub-analysis (or sensitivity analysis) not including non-peer-reviewed publications? Maybe grey literature should be presented in separate. 5) "For question two, only general practice will be of interest...": I really don't see the point of this assumption, as community services are often even more valuable to evaluate the delivery and outcomes of screening interventions. Wouldn't the authors consider them as a subgroup? 6) Please inform if the authors performed a preliminary search in the literature, looking for manuscripts with a similar scope.
--	---

VERSION 1 – AUTHOR RESPONSE

Reviewer 1 comments	Response	Location in main text
This reviewer agrees with the authors' approach of keeping in 'low quality' studies, but it would be helpful to know how they will manage this if there is substantial weakness in the literature for one type of screening or element of delivery.	This is a useful question. For review question 1 (scoping review), where we are collating the components of screening programmes that have been performed in primary care, the quality of the study should have no impact on our findings as we are simply labelling the various identified components. For review question 2 (systematic review), where we explore ways to deliver screening programmes, we will use quality assessment to guide our synthesis of findings, and our interpretation of these. As the reviewer highlights, it may be that the literature is of particularly poor quality in one area; in these circumstances we will need to draw attention to the lack of certainty we have.	"We will, however, use the risk of bias assessment to inform our interpretation of findings for review question two." page 12, paragraph 1. "We will reflect and comment on individual study risk of biases, recurrent types of bias and potential effects of these on the overall conclusions for review question two,

	In response to this comment, we have added two clarifying statements to the manuscript. In the 'Synthesis using best fit frameworks' section we now state "We will, however, use the risk of bias assessment to inform our interpretation of findings for review question two." In the 'meta-biases and confidence in cumulative evidence' section, we have added the clarification "We will reflect and comment on individual study risk of biases, recurrent types of bias and potential effects of these on the overall conclusions for review question two, including how they might create differences in study findings."	including how they might create differences in study findings." Page 12, paragraph 2.
Authors are excluding screening for conditions for secondary or tertiary prevention – this reviewer understands the reason for this (the focus of the protocol is on primary prevention) but I also wonder if any lessons can be learnt from any delivery of these services in primary care.	We agree that there will be important lessons on delivering screening programmes from studies on secondary and tertiary prevention. These programmes are likely to be smaller in scale than for primary prevention and so, in view of the resources required for this study, we are focussing on the larger scale primary prevention programmes in the hope that they will encompass most of the useful lessons about delivery. However, this is an excellent point and we will reflect on this in the discussion section of the final review. We have not made changes to the protocol based on this.	No changes.

Reviewer 2 comments	Response	Location in main text
Intro: maybe some examples of screening programs should be provided (e.g. diabetes, hypertension, echo screening for conditions such as structural heart disease and RHD).	Thank you for this point; we agree that this would improve readers' understanding of context. We have added some examples (such as cervical screening) in the introduction section.	"...such as breast cancer screening or newborn bloodspot testing..." page 1, paragraph 1. "(e.g. cervical screening, general

		health checks)” page 1, paragraph 4.
For research question 2, please clarify if the review is looking for outcomes (e.g. clinical outcomes). If so, they should be better specified, as different types of interventions will be measured by different kinds of outcomes.	Thank you for pointing out something that requires more clarity. The impact of different delivery strategies on clinical outcomes is indeed a worthwhile review question that is lacking in the literature. However, the focus of our review will not be on clinical outcomes. We are interested in any studies that discuss the delivery of screening programmes. As such, on page 4, paragraph 3, we explain “Potential phenomena and outcomes of interest include the systems and strategies used to deliver screening, barriers and facilitators of such approaches, why such approaches were taken...” We originally noted in this sentence that we are also interested to see “...with what impact.” This last point refers to the fact that publications may (but do not need to) report ‘impacts’ in terms of implementation outcomes (e.g. changes in attitudes of staff) or clinical outcomes (e.g. reduced cancer mortality). Due to the variety of possible impacts, and the fact that they are too disparate for meta-analysis, we do not intend to pre-specify those that we are interested in. We instead intend to narratively synthesise the impacts, if they are reported. We recognise the need to make this clearer and the last paragraph of page 4 and the first paragraph of page 5 now reads: “Potential phenomena and outcomes of interest include the systems and strategies used to deliver screening, barriers and facilitators of such approaches, and why such approaches were taken. Some studies will also report impacts of these phenomena and outcomes. These are not essential to be included and could be clinical (e.g. changes in mortality) or implementation	Page 4 paragraph 4. page 5, paragraph 1. Page 11 paragraph 2.

	impacts (e.g. changes in attitudes, reach or maintenance)." We have made extra clarity in our synthesis section in page 11, paragraph 2: "For each review question, this will therefore produce a framework that has been refined by qualitative data and described by quantitative data, which will be narratively synthesised where there is sufficient information."	
In the sentence: "Polygenic risk scores and new directly marketed screening tests, which provide only an estimate of future disease risk and no clear management plan, will be excluded" it is also not clear how the inclusion criteria will be applied: will the authors only include studies with a longitudinal design, that explored outcomes resulting from screening? As an example: echo screening studies, aimed at diagnosing heart disease in high-risk populations, will be excluded? Is that true for research questions 1 and 2 or only for research question 2? Please clarify.	Thank you for raising this question. Although we are only interested in screening programmes that are more than just a test – those that have established ways of managing results – we do not require the studies to be longitudinal. Such programmes are likely to require larger scale coordination, are more likely to create population benefit, and be more likely to be endorsed on a national level, and correspond to modern definitions of screening. We specify that the study should be based on such programmes but the study itself can just be cross-sectional or other non-longitudinal designs. In order to decide on such programmes, we have examined recommendations of national screening committees as well as other programmes that the studies, the author team, or their reading have indicated have clear management plans. We have also discussed these regularly amongst the authorship team in order to achieve consensus. We have described our interest in all study types with primary data in page 5, paragraph 2 under the subtitle 'types of studies': "Empirical studies that involve primary data collection or secondary analyses of primary data will be included. We will include all study designs, whether interventional (including randomised controlled trials and quasi-experimental designs), observational (including cohort studies, case-control studies, cross-sectional studies), qualitative, mixed-methods, case reports or case studies." We hope this provides more clarity and	No changes.

	so we have not made changes to the main text.	
Do the authors plan a sub-analysis (or sensitivity analysis) not including non-peer-reviewed publications? Maybe grey literature should be presented in separate.	We agree that in many circumstances sensitivity analysis can be very important. However, within a best fit framework synthesis, a sensitivity analysis is very difficult to do. A mixed-studies review can take into account the quality and type of publication (peer reviewed, grey literature etc.) when presenting the narrative synthesis and interpretation. As we note within the paper (page 11, paragraph 3): "The authors of the original best fit framework method recommended conducting a sensitivity analysis by removing low quality studies. However, this is not appropriate for this review: assessment of the quality of qualitative studies is often only based on the quality of reporting, and it is widely recognised that for complex interventions even 'low quality' studies may provide useful indications of relevant concepts that need further exploration." We hope that this clarifies our approach and the methodological restrictions within which we are working. We have also addressed the function and management of low quality studies in our response to reviewer 1.	No changes.
"For question two, only general practice will be of interest...": I really don't see the point of this assumption, as community services are often even more valuable to evaluate the delivery and outcomes of screening interventions. Wouldn't the authors consider them as a subgroup?	Thank you for this question: we do recognise that delivery of community services is an important research topic. The exclusion of community services was a pragmatic decision on our part. In conducting such large and complex reviews, we have had to limit the scope to be able to avoid information overload and achieve these within the resources available to us. Community services in particular are very diverse (ranging from pharmacies to community physiotherapists) and we found in early piloting that we quickly became unable to deal with the indicative search findings and screening requirements they would generate. We will ensure that we address this point within the discussion section of the completed review, highlighting it as a	No changes.

	limitation and a priority focus for a future review.	
Please inform if the authors performed a preliminary search in the literature, looking for manuscripts with a similar scope.	We conducted preliminary searches of PROSPERO, the Cochrane database, MEDLINE and CINAHL to look for similar reviews. We also piloted and iteratively developed the search strategy across all included databases. Within these searches, we did not find manuscripts with a similar scope.	No changes.

VERSION 2 – REVIEW

REVIEWER	Nascimento, Bruno Universidade Federal de Minas Gerais, Serviço de Cardiologia e Cirurgia Cardiovascular
REVIEW RETURNED	05-Apr-2021
GENERAL COMMENTS	Dear authors, Thank you for addressing my points and comments. I have no additional suggestions at this point.